# The Influence of Government Effectiveness and Corruption on the High Levels of Homicide in Latin America

**Spencer P. Chainey** [1,*] , **Gonzalo Croci** [1,2] and **Laura Juliana Rodriguez Forero** [1]

1   Department of Security and Crime Science, University College London, 35 Tavistock Square,
    London WC1H 9EZ, UK; gonzalo.croci.18@ucl.ac.uk (G.C.); laura.forero.16@alumni.ucl.ac.uk (L.J.R.F.)
2   Department of International Studies, University ORT Uruguay, Bulevar España 2633,
    11300 Montevideo, Uruguay
*   Correspondence: s.chainey@ucl.ac.uk

**Abstract:** Most research that has examined the international variation in homicide levels has focused on structural variables, with the suggestion that socio-economic development operates as a cure for violence. In Latin America, development has occurred, but high homicide levels remain, suggesting the involvement of other influencing factors. We posit that government effectiveness and corruption control may contribute to explaining the variation in homicide levels, and in particular in the Latin America region. Our results show that social and economic structural variables are useful but are not conclusive in explaining the variation in homicide levels and that the relationship between homicide, government effectiveness, and corruption control was significant and highly pronounced for countries in the Latin American region. The findings highlight the importance of supporting institutions in improving their effectiveness in Latin America so that reductions in homicide (and improvements in citizen security in general) can be achieved.

**Keywords:** homicide; Latin America; government effectiveness; corruption; citizen security

## 1. Introduction

Insecurity is considered to be one of the most important issues facing the Latin America region (World Bank 2015). Nowhere else in the world are homicide numbers or rates of crime as high as in Latin America (Bergman 2018; Briceño-León 2008). High homicide levels also undermine economic and social stability in the region (Bergman 2018) and have significant financial consequences, estimated as being equivalent to six percent of gross domestic product for many countries in the region (World Bank 2019). The homicide problem is also an increasing one: the proportion of homicides across the world that occur in Latin America increased from 29 percent in 2000 to 39 percent in 2017 (Alvarado and Muggah 2018); in recent times, almost all the countries in the region have experienced significant or moderate increases in homicides, and where decreases in homicide have been experienced (e.g., Colombia), these have often been difficult to sustain (Oberwittler 2019; Rennó Santos et al. 2019). (We recognize that Chile is an exception, where homicide rates are below the global average, but use Chile as an example in a later section for examining country heterogeneity and for consistency in the relationships we examine.)

Crime and violence have traditionally been considered symptoms of a country's early stages of development that could be cured with economic growth and reductions in poverty, unemployment, and inequality (Chioda 2017). Supporting this is the notion of homicide rates reducing as part of a pacification process associated with modernization (Pinker 2011). For example, in parallel with development improvements in East Asia, significant declines in homicide rates have been observed (Oberwittler 2019). Over the last decade, countries in Latin America have similarly made progress in their economic and social development, which in turn has contributed to reductions in poverty and inequality (ECLAC 2015; Lustig 2014). Although improvements in poverty rates have stalled in some

countries in Latin America in recent years, in broad terms, citizens in Latin America have experienced real improvements in economic well-being compared to conditions at the end of the 20th Century (Muggah 2017). Citizens in the region have also become healthier and better-educated (Jaitman and Ajzenman 2016), and for the most part, have enjoyed higher levels of democracy, greater citizen participation, and more political transparency (Mainwaring and Pérez-Liñán 2015; O'Donnell 2004). Improvements have also been made in institutional security developments, such as increases in the numbers of police officers and judges operating within the criminal justice system (UNODC 2019). Yet, over the same time period, homicide levels have remained high in Latin America. This situation runs counter to the traditional view of violence as being a symptom of poor human well-being, anomie, and inequality in society (Schultze-Kraft et al. 2018). The situation also indicates that the homicide problem cannot be fixed with the simple addition of more resources, such as increases in the number of police personnel (Soares and Naritomi 2010).

The high levels of homicide experienced in the Latin American region have raised the question of what is it about life in the region that explains why these high levels have persisted when other countries and regions have experienced reductions in homicides and often violence of all types. Studies that have examined the factors to explain the high levels of homicide in Latin America have focused on social inequality, poverty, unemployment, and poor levels of education. To date, very little research has examined how the capacity and competence of government institutions may influence homicide levels. In this paper, we examine the high levels of homicide in the Latin American region by considering how government effectiveness and corruption control may be related to homicide levels. We hypothesize that if governments are ineffective in delivering public services that support an environment where citizens are made secure and the delivery of services are undermined (e.g., by corruption), this then creates an environment in which criminal activity can operate without being checked, of which serious violence in the form of high levels homicides is an outcome.

Our analysis involves two approaches to examining our hypothesis that there is a relationship between government effectiveness, corruption, and homicide. We begin with a qualitative analysis that reviews the main explanations posited by previous studies of the factors that have been suggested to influence homicide levels. From this, we consider how deficiencies of governments in Latin America are an influencing factor on the region's high rates of homicide. Our second approach involves a modest quantitative cross-sectional analysis of the relationship between government effectiveness and homicide, and corruption and homicide. We examine these issues from a regional perspective but discuss experiences from particular countries as examples and to consider country heterogeneity in Latin America. In the final sections of the paper, we discuss the findings and their implications, limitations, and opportunities for further research. The materials and methods we use are described in a later section (to conform with journal article structure).

## 2. Results

### 2.1. Qualitative Review: Factors Explaining High Homicide Levels—A Latin America Focus

Following a long tradition in criminology based on Merton (1968) anomie theory and with a Durkheimian-modernization perspective, most explanations for crime stress the importance of actors and processes within society. According to these views, crime is the result of a chaotic social order resulting from economic disparities, social relations, and transforming norms and values (Cruz 2016; Dicristina 2004). Following from Durkheim's work on the implication that more social development would reduce homicidal dispositions, scholars have argued that variables associated to socio-economic-structural factors are what explain variations in homicide rates because of the way they influence the generation of resentment, feelings of frustration, and the inability to accumulate assets (Chamlin and Cochran 2006; Jacobs and Richardson 2008).

Several studies have indicated that high levels of homicides are associated with social inequality and unequal life chances for citizens (Avison and Loring 1986; Bourguignon

et al. 2003; Chamlin and Cochran 2006; Hansmann and Quigley 1982; Imbusch et al. 2011; Pratt and Godsey 2003; Trent and Pridemore 2012), economic development and deprivation (Chamlin and Cochran 2005; LaFree 1999; Messner 1989; Nivette 2011; Wilkinson 2004), and inequalities in income (Briceño-León et al. 2008; Fajnzylber et al. 2002; Nadanovsky and Cunha-Cruz 2009). However, over the last twenty years, there have been many improvements in inequality and income distribution across Latin America (Ocampo and Vallejo 2012). Over this period, the Gini coefficient (a measure of inequality) reduced by almost ten percent from 0.542 to 0.486 in the region, yet homicide rates increased (Bergman 2018; Vilalta et al. 2016). Examined more specifically in relation to certain countries in the region, Brazil and Mexico have experienced periods of economic growth and decreasing income inequality, but high levels of homicide have persisted. In other countries such as Paraguay and Bolivia, inequality has persisted at high levels, but homicide levels have remained relatively low in comparison to other countries in the region. These observations suggest that a simple inequality–homicide relationship is not consistent in Latin America.

According to Becker (1968), individuals become criminals because of the financial benefits and rewards that are available from criminal activity in comparison to the risk of apprehension and the severity of punishment. From this hypothesis, scholars have posited that economic incentives are key factors to consider when explaining illegal activity and the violence associated with it (Jaitman and Machin 2016; LaFree 1999; Nivette 2011; Wilkinson 2004). From this, researchers have worked on the hypothesis that high levels of poverty are a key factor associated with high homicide rates (Baumer and Wolff 2014; Cerqueira and Lobão 2004; Chon 2011; Hsieh and Pugh 1993; Koeppel et al. 2015; Messner and Zimmerman 2012; Pridemore 2008). Other studies, such as Rennó Rennó Santos et al. (2018), have, however, questioned the poverty and homicide relationship, showing that poverty was only related to homicide in countries with low homicide rates. Since 2005, the Latin America region has experienced its most rapid period of economic growth since the 1960s (Bustillo and Velloso 2015), GDP per capita grew from USD 3547 in 2003 to USD 10,278 in 2014 (Bergman 2018), and the proportion of the region's 632 million residents in poverty declined from 41.7 percent to 25.3 percent (Muggah 2017). Additionally, it is not typically the poorest countries nor is it the poorest areas within countries in the Latin America region that experience the highest homicide rates (Briceño-León 2008). Collectively, these findings suggest that high levels of poverty do not inevitably lead to high homicide rates.

Unemployment has been identified by many researchers as a variable influencing crime and violence. In a similar manner to Becker (1968) rationale, limited employment opportunities increase the temptation to seek financial rewards from illegal activity. The reduction of legitimate labor market opportunities can make criminal activity more attractive, with high levels of unemployment contributing to feelings of frustration, which in turn results in higher levels of violence (Bergman 2009; Buvinic et al. 1999). Youth unemployment, in particular, is the key factor considered as having an influencing factor on violence (Chioda 2017) because youths are especially susceptible to predation and criminal behavior, and they consider the financial benefits of engagement in criminal activity to be greater than the benefits offered in the formal labor market. In addition to seeking material gain, violent competition between young people, males in particular, is associated with them seeking influence and honor (Eisner 2008). Moreover, youth unemployment is strongly related to gang recruitment and membership (Decker and Van Winkle 1996; Pitts 2007), with violence often associated with these gang activities in the form of territory protection, retaliation, and revenge (Brown and Osterman 2012; Eisner 2003). Others have, however, stressed that employment alone may be insufficient to deter involvement in crime, and that the quality of employment is a more relevant factor to prevent individuals from entering into criminal activity (Freeman 1995; Levitt 2001; Muggah 2017). Several other studies have gone further and questioned the link between unemployment and homicide (LaFree 1999; Raphael and Winter-Ebmer 2001). For instance, between 2002 and 2014, the unemployment rate in Latin America reduced from 11.2 percent to 6.0 percent, and young male unemployment (ages 15–24) reduced from 15.7 percent to 11.8 percent (World Bank

2018). In addition, labor formalization and social security have increased substantially throughout the Latin America region over the same time (Cecchini et al. 2015).

Education is another factor that has been commonly associated with crime and violence. Researchers have found that an increase in school attendance and completion can reduce crime (Muggah 2017; Rivera 2016), that schooling significantly reduces the probability of incarceration and arrest (Lochner and Moretti 2004), and that improving education can yield significant social benefits and act as a key policy tool in efforts to reduce crime (Machin et al. 2011). However, other studies have shown weaknesses in a simple education and crime relationship, especially for homicide. For example, Heinemann and Verner (2006) showed that the average number of years of schooling was not related to rates of homicide. In recent decades, countries in Latin America have experienced considerable improvements in education levels. Between 2004 and 2017, the region's secondary school enrollment grew from 85 percent to 95 percent, and tertiary enrollment from 29 percent to 50.6 percent (World Bank 2018). Government expenditure on education across the region also increased from 3.7 percent of GDP in 2004 to 5.2 percent in 2014 (World Bank 2018). Brazil in particular has experienced substantial improvements in its education measure; however, levels of homicide increased over the same period. In contrast, improvements in education in Peru have stagnated in recent years, yet rates of homicide have remained relatively low.

To date, most cross-national studies that have examined variations in homicides have only included a small representation of countries from Latin America (Rennó Santos et al. 2018) and have primarily included countries from North America, Europe, and other developed nations. Developed countries in other parts of the world have homicide rates that are much lower than those in Latin American countries (Stamatel 2009), meaning that most cross-national homicide studies have included data for countries that feature at the lower end of the global scale of homicide rates. Other cross-national studies of homicide have examined the influence of rapid and disorganized urbanization (Neumayer 2003), population density (Nivette 2011), age structure (Rennó Santos et al. 2019), transition to democratic rule (Neumayer 2003; Rivera 2016), cultural masculinity (Neapolitan 1994), and trafficking of firearms and organized crime (Esparza et al. 2019; Garzón-Vergara 2016). Although the results from these studies and the many others that have been referred to in this qualitative review have provided some important insights into trends in homicide in Latin America, collectively these studies have often only offered modest explanatory value or have failed to offer consistent explanations for the high levels of homicide across the region. In addition, the small samples of Latin American countries that have been included in many cross-national studies of homicide could mean that their results are less valid to the Latin American context.

Some researchers have begun to emphasize a link between the roles performed by government institutions and the international variations in social and economic conditions and how institutional weakness can have far-reaching implications in society. Research examining political institutions has begun to show how issues of institutional strength, stability, legitimacy, authority, and effectiveness, rather than just formal institutional volume and design, influence the expectations of individuals and their behaviors and the social and economic conditions that result (De Boer and Bosetti 2015; Lafree and Tseloni 2006). Many of these findings have drawn from the field of political science and economics that has increasingly suggested the presence of a relationship between poor governance and underdevelopment (Acemoglu and Robinson 2012; Kaufmann et al. 2009; Levitsky and Murillo 2009; World Bank 2011). This includes Dahlstrom and Lapuente (2017), who have examined how good governance improves public policy and practice. They argue that governments that are meritocratic experience higher levels of government effectiveness, lower levels of corruption, and are better placed in adopting modernizing reforms. From this, when there is a structure within government institutions that allows for a productive relationship between politicians and civil servants, this increases efficiency, improves public policy and practice, and minimizes corruption (Dahlstrom and Lapuente 2017).

The idea that variations in the level of crime are associated with variations in government effectiveness and corruption has, however, received little attention, although some researchers have begun to suggest a link. Stamatel (2009, pp. 17), for example, recommended that research on patterns of violence must start to examine "regime types, efficacy and legitimacy of governments" and incorporate "political factors, especially the role of the state in maintaining law and order". This has led to the suggestion that factors associated with governance and institutional capacity should be considered in models for examining relationships with crime (Nivette and Eisner 2013). This has begun to happen, with several studies showing how ineffective institutional structures can have an influence on crime (Azfar and Gurgur 2005; Huebert and Brown 2019; Nivette and Eisner 2013; Pinker 2011; Wenmann and Muggah 2010). With particular consideration to the Latin American region, Neumayer (2003) has suggested that political governance could be a predictive variable for explaining the variation in homicide rates across the region. Building on the suggestions from political science and economics on the presence of a relationship between poor governance and underdevelopment and how institutional effectiveness can influence the behavior of individuals, we argue that in the context of citizen security, the variation in good governance may also result in variations in levels of crime.

The recency of these considerations about how the effectiveness of government institutions may influence homicide rates means that the theoretical concepts to support these arguments are under-developed. Bergman (2018) and Rivera (2016) have argued, however, that when institutions are weak, when state authorities have limited capacity, and when police agencies and criminal justice systems are ineffective, major opportunities for illegal activity can appear. We add to this by arguing that when government authorities are ineffective in providing adequate services that offer security to citizens, it can create a void in which criminal activity has the potential to thrive. We argue that institutional effectiveness can influence the criminal behavior of individuals, and the act of homicide is the most extreme of this criminal behavior. Socio-economic structural factors do have a role to play in helping to understand the variation in homicide levels but fail to explain the full extent of violence in Latin America. We argue that institutional strength and effectiveness have an effect on homicide rates, and in particular could be a key factor for explaining the high homicide rates experienced in the Latin America region. We expand on these theoretical concepts that relate to institutional effectiveness and homicide rates in the discussion section.

In addition to institutional effectiveness, we argue that corruption plays an influencing role within this relationship with homicide rates. Although corruption is not traditionally thought of as a homicide issue, evidence indicates that a low level of corruption is critical to a peaceful society, high levels of corruption undermine state institutions and democracy (Dahlstrom and Lapuente 2017), and corruption reduces the legitimacy of the justice system and increases impunity (Bergman 2018) and contributes to insecurity (Azaola 2009; Frühling 2009; Institute for Economics and Peace 2018). Oberwittler (2019, p. 32) also states that "it seems surprising that corruption has rarely been tested as a predictor of homicide, considering the role of corruption in shady business practices and dysfunctional governance". In Latin America, despite improvements in social and economic development, levels of corruption remain high. We hence also argue that institutions that are ineffective in controlling for corruption are likely to experience higher levels of homicide. This is because of the undermining effect that corruption can have in the operation of good governance, public policy, and practice, which in the context of citizen security can weaken the provision of state services for controlling crime, reduces the legitimacy of the justice system, and leads offenders to believe that criminal behavior will go unpunished.

Following our qualitative analysis, we perform a modest quantitative analysis in which we test three hypotheses: countries with lower levels of government effectiveness are more likely to experience higher levels of homicide; countries with lower levels of corruption control are more likely to experience higher levels of homicide; countries from the Latin American region are more likely to exhibit lower levels of government

effectiveness and higher levels of corruption, and in turn are more strongly correlated for these factors against homicide than other parts of the world.

## 2.2. Statistical Analysis

Table 1 lists the descriptive statistics of the variables used in the analysis (see the Materials, Methods, and Limitations section for an explanation for the choice of these variables). Homicide rates varied between 0.1 per 100,000 population to 92.7 across the data sample. Each of the other variables were considered to provide a representative spread of values, albeit while including a large representation of Latin American countries in the data sample.

**Table 1.** Descriptive summary of each variable (*n* = 39).

| Variable | Mean | Standard Deviation | Minimum | Maximum |
|---|---|---|---|---|
| Homicide (rate per 100,000) | 13.38 | 19.0 | 0.10 | 92.70 |
| Natural log of homicide | 0.69 | 0.69 | −1.00 | 1.97 |
| Government effectiveness | 0.53 | 0.95 | −1.15 | 2.23 |
| Control of corruption | 0.42 | 1.13 | −1.26 | 2.32 |
| Inequality | 11.64 | 8.39 | 2.50 | 42.30 |
| Unemployment | 7.89 | 4.84 | 1.46 | 24.79 |

Table 2 presents a correlation matrix that shows that government effectiveness and control of corruption were correlated (as expected and discussed in the Methods and Materials section, hence the decision to use these variables in separate models). Income inequality was significantly related to government effectiveness and corruption control, with this relationship being examined further in each model and involving tests for multi-collinearity. No significant relationships were found between unemployment and any of the other variables.

**Table 2.** Correlation coefficient matrix of independent variables of cross-national analysis (*n* = 39). *** *p* < 0.001.

| | Government Effectiveness | Control Corruption | Income Inequality | Unemployment |
|---|---|---|---|---|
| Government effectiveness | | | | |
| Control corruption | 0.957 *** | | | |
| Income inequality | −0.606 *** | −0.588 *** | | |
| Unemployment | 0.255 | 0.194 | −0.245 | |

Tests for multicollinearity that were applied to Model 1 showed that for each variable, the variance inflation factor (VIF) was less than five, suggesting that multicollinearity was not present in the model. The results from Model 1 identified government effectiveness and income inequality to be significantly related to homicide, and in the expected directions: as government effectiveness decreased, homicide rates increased, and as income inequality increased, homicide rates increased (Table 3). Unemployment was not significantly related to homicide. Similarly, in Model 2, controls for corruption and income inequality were significantly related to homicide, and unemployment was not significant. Additional test results showed that in Model 2, the VIF for each variable was less than five, suggesting that multicollinearity was not present in the model.

**Table 3.** Regression analysis results for homicide, government effectiveness, controls for corruption, income inequality, and unemployment (*n* = 39). * *p* < 0.05, ** *p* < 0.01, *** *p* < 0.001.

|  | Model 1 | Model 2 | Model 3 | Model 4 |
|---|---|---|---|---|
|  | *β* | *β* | *β* | *β* |
| Intercept | 0.540 * | 0.449 * | 0.402 * | 0.315 |
| Government effectiveness | −0.418 *** | - | −0.345 *** | - |
| Control of corruption | - | −0.307 *** | - | −0.261 *** |
| Inequality | 0.029 *** | 0.032 *** | 0.016 | 0.013 |
| Unemployment | 0.004 | −0.001 | 0.012 | 0.011 |
| Latin America | - | - | 0.423 * | 0.572 ** |
| Adjusted R$^2$ | 0.67 | 0.63 | 0.68 | 0.67 |
| F statistic | 24.05 *** | 20.10 *** | 20.45 *** | 19.90 *** |
| Log likelihood | −18.84 | −21.13 | −16.76 | −17.14 |

The second tests involved repeating Models 1 and 2 but with the inclusion of the regional dummy variable of Latin America in each model to identify if the model results changed anyway. In each model, the VIF for each variable was less than five, suggesting multicollinearity was not present in the models. In Model 3, government effectiveness was found to be significantly related to homicide rates, but income inequality was no longer significant (see Table 2). Unemployment remained not significant. The coefficient for government effectiveness indicated that for every 0.1 index unit increase in government effectiveness, the homicide rate would reduce by about three percent. Model 3 also indicated a significant regional effect from Latin America, shown by the significant positive relationship between the countries within Latin America and homicide rates (β = 0.423, *p* < 0.05). Similarly, once the Latin America dummy variable had been included into Model 4, controls for corruption continued to be significantly correlated to homicide rates, income inequality and unemployment were not, and a significant regional effect was present from Latin American countries. The coefficient for control of corruption indicated that for every 0.1 index unit increase in control of corruption, the homicide rate would reduce by about three percent. Collectively, the results from models 3 and 4 showed that homicide rates were significantly higher in countries where government effectiveness and controls for corruption were poor, and that this influencing factor was most significantly apparent in Latin American countries. Additionally, once the regional dummy variable of Latin America had been introduced into the models, income inequality was no longer significantly related to homicide. On average, a country in Latin America was associated with a 53 percent higher level of homicide and a 77 percent higher level of homicide when, respectively, government effectiveness and control for corruption were taken into consideration.

## 3. Discussion

To date, most research that has attempted to explain variations in homicide rates has used socio-economic structural variables such as inequality, poverty, and unemployment. Many of these studies have included some countries from Latin America to examine the international variations in homicides or have only exclusively studied countries from Latin America when seeking to examine the influence of structural variables. Although these studies have proved useful in illustrating the role of socio-economic factors in helping to understand the variation in homicide levels, the results have been inconsistent in explaining the full extent of homicide in Latin America. The Latin American region has the highest homicides rates in the world. In recent decades, overall living conditions have improved in the region, yet high homicide levels have persisted. Improvements to living conditions, social standards, and economic growth have long been considered as cures to violence. This appears to have not been the case in Latin America, and consequently has raised the

question of what it is about life in the region that makes homicide (and violence in general) so prevalent. The inconsistency in the relationship between traditional socio-economic structural variables and homicide in Latin America also raises questions about the extent to which violence can be considered as a symptom of a country's early stages of development and homicide rates reduce as part of a pacification process associated with modernization. It seems likely, particularly in the Latin American context, that other factors appear to be relevant in this homicide–development–modernization relationship.

The research on homicide in Latin America has yet to clearly identify what it is about life in Latin America that results in high homicide levels being so prevalent. This study took the approach of examining how factors associated with the deficiencies of government influence the homicide levels in the region. In this study, we began by posing the hypotheses that two variables in particular—ineffective government and poor controls for corruption—have an influence on homicide levels. The study included a large sample of Latin American countries to determine if these two governance variables were particularly prevalent in helping to explain high homicide levels in Latin America.

Our results suggest a relationship is apparent between government effectiveness and homicide levels, and corruption control and homicide levels. As previously stated, because of the recency of arguments suggesting these types of associations, theoretical concepts that may explain this relationship are underdeveloped. Building on the suggested presence of a relationship between poor governance and underdevelopment that has emerged from political science and economics, here we extend this argument by suggesting an institutional organizational theoretical concept that relates to citizen security, in particular homicide. The appropriate use of power and the adequate allocation of public resources in society provide a fundamental basis for the provision of citizen security. When governments fail to allocate sufficient resources to the public institutions responsible for social control and physical infrastructure, and fail to invest in the formation, development, maintenance, and functionality of these public entities, their effectiveness in providing citizen security is hindered. In a public service provision system where corruption is present, particularly in relation to law enforcement and the judicial system, this can lead to high levels of perceived lack of justice and the legitimization of violence. Moreover, when people within public institutions (such as the police) find themselves surrounded by corruption, they may have to accept and even participate in corruption despite their values (Rose-Ackerman 2001). An environment within which there is tolerance of corruption may result in it perpetuating as a cultural norm (Warburton 2001) and undermine institutional efforts to improve citizen security. Collectively, rather than or in addition to economic and social development contributing to homicide reduction, how institutions operate and how effective they are in the provision of citizen security can influence the homicide levels that are observed.

Our findings are illustrated further in Figures 1 and 2 and illustrate the prevalence in Latin America of the government effectiveness and homicide, and corruption control and homicide, relationships. In Figure 1, high homicide rates in countries in Latin America correlate strongly with the high negative values for government effectiveness. In contrast, non-Latin American countries that experienced low homicide rates correlate with positive scores for government effectiveness. Figure 1 shows country heterogeneity in the Latin American region (e.g., in Chile, where low homicide rates are matched with a positive score for government effectiveness); however, the overall relationship is apparent—higher rates of homicide are related to lower levels of government effectiveness. Similarly, in Figure 2, high homicide rates in Latin American countries are shown to correlate strongly with the high negative values for control for corruption in comparison to non-Latin American countries, where low homicide rates correlate with positive scores for control of corruption. Differences in controls of corruption are apparent in Latin American countries, but this country heterogeneity again matches with the relationship between the homicide levels that each country experiences. For example, in Mexico, the homicide rate was 21.3 and the control for corruption value was −0.41 compared to Chile, where the homicide rate was 2.5 and the control for corruption value was 1.57.

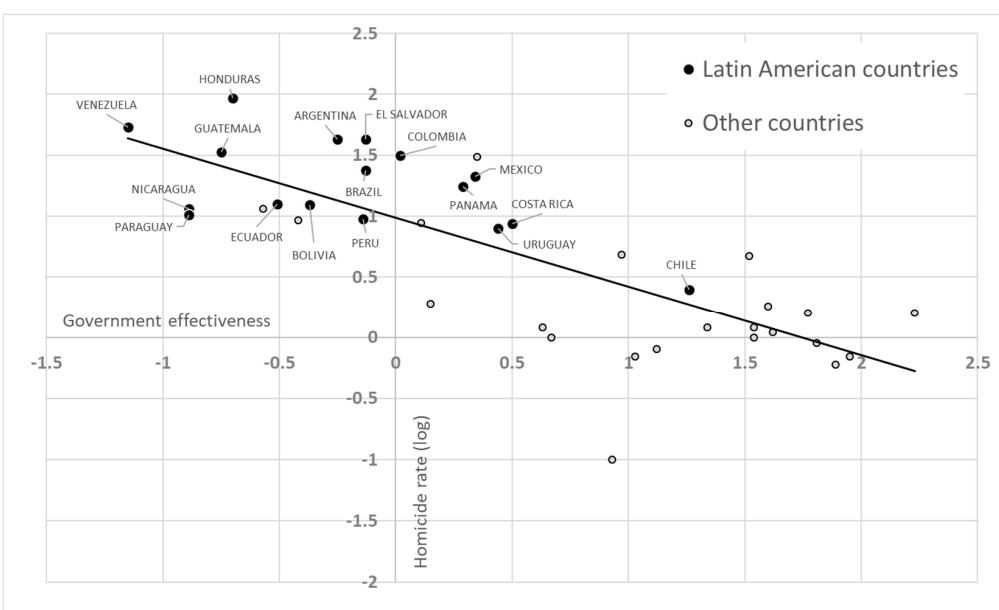

**Figure 1.** Government effectiveness and homicide rates.

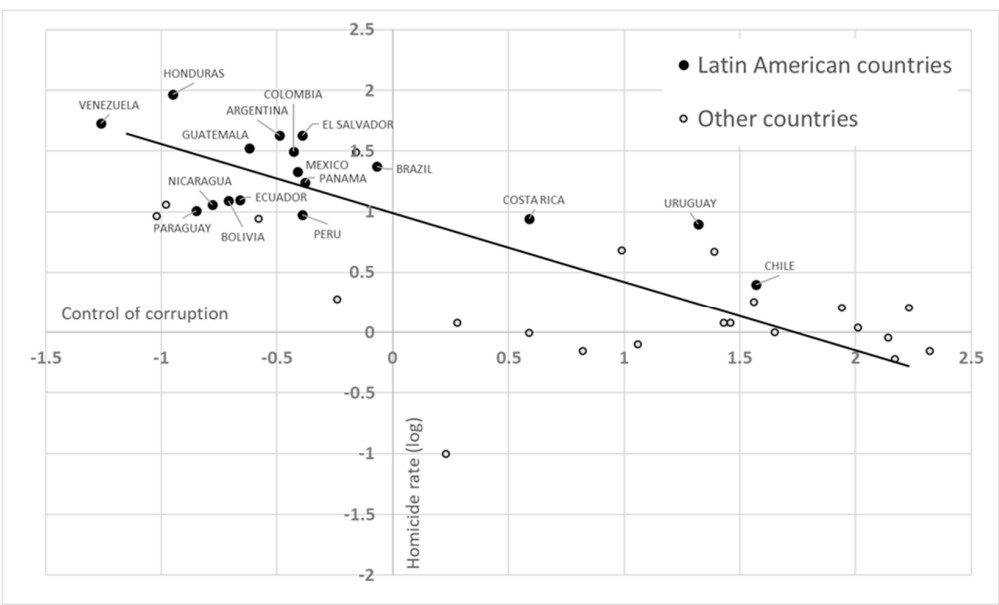

**Figure 2.** Control of corruption and homicide rates.

The inclusion of the variables of unemployment and inequality in the models, alongside the dummy variable for Latin America and the variables for government effectiveness and corruption control helped to examine the particular influence of these factors in explaining the variation in homicide in the Latin American context. These results initially showed that income inequality, alongside government effectiveness and corruption control, was a significant influencing factor for explaining variations in homicide for the full international sample of countries, but that unemployment was not an influencing factor. However, when the dummy variable for Latin America was included in the models, the variable for income inequality was no longer significant. This leads to the suggestion that the relationship between income inequality and homicides, once government effectiveness and controls for corruption have been considered, is not as pronounced in Latin America as previous studies have indicated. Although we do not go so far to claim that income inequality should be discounted as an influencing factor on homicide in the Latin American region, the results from the current study may suggest that inequality is an outcome related to the

deficiencies of government, with high homicide levels being an outcome if governments are ineffective in addressing inequality. In addition, the relationships between government effectiveness and homicide, and controls for corruption and homicide, also appear to be more consistent between countries in the Latin American region than the structural variables we review in the current study.

Adding to our institutional organizational concept associated with citizen security, our findings lead us to suggest that if government authorities are ineffective in providing adequate services that offer security to citizens, it can create a void in which criminal activity has the potential to thrive. Similar to suggestions from political science and economics of the presence of a relationship between poor governance and underdevelopment and how institutional effectiveness can influence the behavior of individuals, in a citizen security context, institutional effectiveness can influence the criminal behavior of individuals, of which the act of homicide is the most extreme example. We then believe that corruption has a further undermining effect that influences this relationship between governance and homicide because of how the presence of corruption weakens the provision of state services for controlling crime, reduces the legitimacy of the justice system, and leads offenders to believe that a homicide they commit will go unpunished. These factors (government effectiveness and corruption) are most prevalent in the Latin America region and, we suggest, contribute to explaining why high homicide levels persist in the region. In countries in Latin America where there are more effective government institutions and more effective controls for corruption (e.g., Chile), homicide rates are lower. Although our research did not examine the specific effectiveness of police services and the criminal justice system directly, our findings offer a suggestion that when governments are ineffective in providing services that enforce the law, prevent crime, and are ineffective in delivering criminal justice, high homicide levels result. We recognize that the mechanisms that connect government effectiveness and corruption to homicide require further enquiry but suggest that when government institutions are seen to be weak, high levels of homicide are likely to be present, particularly when corruption is rife, and individuals consider there to be significant gains from criminal enterprise.

In the current study, our aim has been to review, examine, and then qualify the influence of government effectiveness and corruption control on homicide levels and to do so by building on previous research that has begun to suggest the possibility of such a relationship. The results from the study suggest that a relationship is apparent between the variation in homicide levels, government effectiveness, and corruption control, and indicates that these, perhaps, are key features that help explain why homicides are so high in the Latin America region. Our statistical analysis is modest, and we hope it helps to initiate further research that examines this relationship in more detail, particularly and specifically regarding the law enforcement and criminal justice environment and in the examination of homicide in other developing regions such as Africa. Our statistical analysis examined a single snapshot in time. We recommend more detailed analysis that examines if the relationship between changes in homicide levels over time continues to correlate with our modest cross-sectional analysis findings. Additionally, further research should also examine if rapid escalations in violence in countries in Latin America (and elsewhere) are directly related to severe weaknesses in government effectiveness and corruption control. Countries in Latin America also have high levels of impunity, a specific factor that we hypothesize is also directly related to government ineffectiveness and poor corruption control in a policing and criminal justice environment. Research that examines impunity specifically as a variable and its relationship with national homicide levels would also be worthy of study.

We also encourage further research that examines relationships between explanatory variables at different levels or scales. Our study provides a macro-level examination of the conditions we test for their influence on homicide levels. The challenge, then, is to explain how these macro-level explanations translate to micro-level contexts of homicide commission. This requires analysis of variables that explain homicide levels to be examined

for their consistency at the state, city, and neighborhood levels. In this vein, we suggest the findings from the current study may offer some help in translating macro-phenomena to micro-activity. That is, where there exist conditions that are conducive to crime, the void that is left by the state because of the ineffective formulation and implementation of solid policies and the ineffective provision of public services (which can be undermined further by corruption) can be exploited for criminal endeavor. If government effectiveness is found to be consistent as an explanatory variable for the variation in homicides across all geographic scales, this could translate into identifying interventions that are most effective in decreasing homicides that are relevant to that scale (i.e., national policies, city-wide programs, micro-place interventions, and individual-specific programs). Increasingly, what works and what does not work in homicide reduction, and that is sensitive to the setting, is being documented (Cano and Rojido 2016; Chioda 2017; WHO 2014); hence, there is the opportunity to draw from this practice and match it to the explanations for the variation in homicide levels, relevant to the context and scale of the intervention.

## 4. Materials, Methods, and Limitations

An extensive literature review was conducted of studies about homicide and its relationship with social and economic structural factors. This review included identifying theoretical principles that supported these relationships. The literature review focused on studies published since 1995, and in particular, studies that included countries from Latin America in the analysis. The qualitative review then examined experiences from Latin American countries that related to the social and economic structural factors that researchers had posited as being correlated to homicide. The qualitative review included studies that reported changes in these variables over time in certain Latin American countries, so in places we comment on these changes in relation to the changes in homicide that these countries have experienced. The qualitative review then examined the published, albeit limited, research about how the effectiveness of government institutions and corruption may influence homicide rates, from which we pose some theoretical concepts that may explain the presence of a relationship.

The quantitative analysis involved a modest cross-sectional regression analysis of the relationship between homicide and several variables for an international sample of countries, but with specific attention paid to countries in Latin America. Thirty-nine countries were included in the study (see Table A1), 17 of which were from Latin America. The 22 non-Latin American countries were chosen because of data availability for these countries and because each has been extensively researched and included in cross-national studies on changes in crime (Van Dijk et al. 2012; Tilley et al. 2011). To our knowledge, very few or no other studies that have examined international variations of homicide have used a sample consisting of such a large representation of Latin American countries. Our full sample size is consistent with most other studies we reference in the Results section.

Data on homicide rates were obtained from the World Bank DataBank (World Bank 2018). These data are derived from the United Nations Office on Drugs and Crime's (UNODC) International Homicide Statistics database. UNODC homicide data draws on both criminal justice data and World Health Organization health data in its creation and is considered to be the most reliable for cross-national study (Oberwittler 2019). These data refer to intentional homicides and are defined as the "unlawful homicides purposely inflicted as a result of domestic disputes, interpersonal violence, violent conflicts over land resources, inter-gang violence over turf or control, and predatory violence and killing by armed groups" (World Bank 2018). Data for 2014 were used, as these were the most recent, most complete, and matched with the year of the other variables for the countries examined. Natural log transformations were applied to the homicide rate data to address skewness and to support how the results were presented. We resisted conducting a panel analysis using several years of data, primarily so that we could first test our hypotheses, consider the results from a simpler cross-sectional study, and create a foundation from which further research could develop.

Data for 2014 on government effectiveness and control of corruption were extracted from the World Bank Worldwide Governance Indicators (WGI) project dataset (World Bank 2017). The WGI dataset was chosen, as it is considered to be a reliable and comprehensive dataset for examining cross-national variations in governance and its relationship with crime (Azfar and Gurgur 2005; Kaufmann et al. 2010). Government effectiveness is defined as the capacity to formulate and implement sound policies and deliver public services. Control of corruption is defined as the extent to which public power is exercised for private gain, and the 'capture' of the state by elites and private interests (Kaufmann et al. 2010). Each measure is provided as a standardized score ranging from −2.5 to 2.5. Lower scores represent lower levels of governance. WGI data on rule of law were considered as another variable to examine but were not used, as this measure includes within it data on homicide rates, thus rendering issues of tautology if used. We reflect on the rule of law (and impunity) in the discussion section.

Inequality and unemployment were selected as variables to include in the analysis because of their extensive use in previous studies and their suggested relevance to the Latin American context. Data on inequality were extracted from the United United Nations (2018) for the year 2014 and represent the ratio of the average income of the richest 20 percent to the poorest 20 percent. Unemployment data for 2014 were obtained from the International Labor International Labor Organization (2018), ILOSTAT database, and represent the proportion of the labor force that is unemployed but available for and seeking employment. A Latin America regional dummy variable was created to determine if the influences of variables on homicide were greater in Latin American countries than in non-Latin American countries.

The quantitative analysis was organized into two tests. The first test involved two separate multivariate models: a model that examined the relationship between homicide, government effectiveness, inequality, and unemployment (Model 1), and a model that examined the relationship between homicide, control of corruption, inequality, and unemployment (Model 2). We created two separate models out of concern for multicollinearity between government effectiveness and corruption control. The WGI dimensions of governance are created from a number of the same sources, so it is likely that government effectiveness and corruption control from the source we used are correlated to each other. A preliminary model that included government effectiveness and corruption control with the other structural variables confirmed this with variation inflator factor results for government effectiveness and corruption control that were greater than 7.5. The second tests (Models 3 and 4) involved replicating Models 1 and 2 but for each including the dummy variable to determine if any influencing factors were particularly apparent in Latin American countries. VIF values for each variable in Models 1 to 4 were less than five, suggesting that multicollinearity was not present.

An additional series of tests was performed using a Monte Carlo simulation to examine the reliability of the results. WGI indicators for government effectiveness and control of corruption are provided with data on their standard errors. The Monte Carlo simulation used the index scores and their standard errors to generate a range of estimates for each indicator, using these estimates to repeat the regression models and compare results to those from the previous models. The Monte Carlo simulation tests showed there was very little difference in the results from the simulated and empirical models, suggesting the results from the empirical models were reliable. To comply with article length, the Monte Carlo simulation results are not included, but are available from the authors.

The study used data from the World Bank WGI dataset for government effectiveness and corruption control. These data are based on a compilation of administrative data and surveys from disparate sources. When using survey data, a concern is their subjectivity, and the extent to which opinions and perceptions adequately capture reality. Kaufmann et al. (2010) have previously cited these limitations and have addressed the use of the WGI data as being adequate (Kaufmann et al. 2004, 2007, 2010). We do, however, highlight in particular that the WGI data include information from survey results. Although opinion

and perception data do come with pitfalls, they are considered to be a valuable source of information for assessing governance itself and more complex issues within the context of governance, such as government effectiveness and corruption (Jong-Sung and Khagram 2005). Opinions and perceptions matter, as government entities design their policies and evaluate their impact based on the experience, views, feelings, and impressions of citizens. This is also true for social dynamics such as corruption, in that the experience of, views about, and awareness of corruption by citizens influences their opinions and perceptions about what measures are in place for controlling corruption and hence the extent of corruption that may be present in society. Additionally, there are few alternatives to the data used in the current study for variables such as corruption because of its hidden and unrecorded nature, and in terms of governance, any fact-based measures or statistics that do exist do not necessarily portray the real situation. However, we are aiming to use alternative measures of corruption in future research to examine if both government effectiveness (from the current source) and an alternative measure of corruption can be used in the same model without issues of multicollinearity.

The current study identified the significant relationship that was associated with homicide rates and the Latin America dummy variable. This confirmed our original inspiration for the research that sought to examine why the Latin America region has such high levels of violence compared to other parts of the world. Our findings showed that although government effectiveness and corruption control appear to help explain why countries in Latin America experience high homicide levels in comparison to other countries across the world, the variables on their own do not provide a full explanation. This is to be expected because of the complexity and multi-dimensional aspects of homicide. This, therefore, requires the current study to be replicated by examining other dimensions of governance and institutional competence that may influence homicide rates, such as impunity, gun control polices and gun ownership, weapons trafficking, the influence of illegal markets, and the use of alternative measures for government effectiveness and corruption control.

## 5. Conclusions

The Latin American region experiences the highest levels of homicide in the world. To date, most cross-national studies of homicide have examined its relationship to socio-economic structural variables. Although these findings have been useful on an international level, they have not fully explained why the Latin American region stands out as experiencing persistently high levels of homicide. Using an international sample, government effectiveness and control of corruption were found to be significantly associated with homicide rates. In countries with high homicide rates, there were low levels of government effectiveness and poor controls for corruption. Of particular note was that the relationship between homicide levels, government effectiveness, and corruption control was most apparent for countries in Latin America. These findings indicate that the ineffectiveness of governments and poor controls for corruption appear to play a contributing role in the homicide levels experienced in Latin American countries. From this, we introduce an institutional organizational concept associated to citizen security, suggesting that if the state is absent in providing adequate services that offer security to citizens, it can create a void in which criminal activity has the potential to thrive. Corruption then further undermines the provision of services that offer security to citizens, with serious violence in the form of high levels of homicide being a consequence. Although the study of socio-economic structural variables in relation to homicide has been useful, development programs that improve the effectiveness of government services and improve controls for corruption could offer a significant contribution to addressing the high levels of homicide experienced in the Latin American region.

**Author Contributions:** Conceptualization, S.P.C.; methodology, S.P.C. and L.J.R.F.; software, S.P.C.; validation, S.P.C., G.C. and L.J.R.F.; formal analysis, S.P.C., G.C. and L.J.R.F.; investigation, S.P.C.; resources, S.P.C.; writing—original draft preparation, S.P.C., G.C. and L.J.R.F.; writing—review and editing, S.P.C., G.C. and L.J.R.F.; visualization, S.P.C.; supervision, S.P.C.; project administration, S.P.C. All authors have read and agreed to the published version of the manuscript.

**Funding:** This research received no external funding.

**Institutional Review Board Statement:** Not applicable.

**Informed Consent Statement:** Not applicable.

**Data Availability Statement:** Data are provided in Table A1.

**Conflicts of Interest:** The authors declare no conflict of interest.

## Appendix A

**Table A1.** Countries included in the research, homicide rates, and WGI measures for government effectiveness and corruption control.

| Country | Homicide Rate 2014 (per 100,000) | Government Effectiveness | Control of Corruption |
|---|---|---|---|
| Argentina | 42.7 | −0.25 | −0.49 |
| Bolivia | 12.4 | −0.37 | −0.71 |
| Brazil | 23.8 | −0.13 | −0.07 |
| Chile | 2.5 | 1.26 | 1.57 |
| Colombia | 31.3 | 0.02 | −0.43 |
| Costa Rica | 8.7 | 0.50 | 0.59 |
| Ecuador | 12.5 | −0.51 | −0.66 |
| El Salvador | 42.7 | −0.13 | −0.39 |
| Guatemala | 33.5 | −0.75 | −0.62 |
| Honduras | 92.7 | −0.70 | −0.95 |
| Mexico | 21.3 | 0.34 | −0.41 |
| Nicaragua | 11.5 | −0.89 | −0.78 |
| Panama | 17.5 | 0.29 | −0.38 |
| Paraguay | 10.2 | −0.89 | −0.85 |
| Peru | 9.5 | −0.14 | −0.39 |
| Uruguay | 7.9 | 0.44 | 1.32 |
| Venezuela | 53.8 | −1.15 | −1.26 |
| Australia | 1.1 | 1.62 | 2.01 |
| Belgium | 1.8 | 1.60 | 1.56 |
| Bulgaria | 1.9 | 0.15 | −0.24 |
| Canada | 1.6 | 1.77 | 1.94 |
| Czech Republic | 0.1 | 0.93 | 0.23 |
| Estonia | 4.8 | 0.97 | 0.99 |
| Finland | 1.6 | 2.23 | 2.23 |
| France | 1.2 | 1.34 | 1.43 |
| Hungary | 1.2 | 0.63 | 0.28 |

**Table A1.** *Cont.*

| Country | Homicide Rate 2014 (per 100,000) | Government Effectiveness | Control of Corruption |
|---|---|---|---|
| Ireland | 1.2 | 1.54 | 1.46 |
| Netherlands | 0.9 | 1.81 | 2.14 |
| Philippines | 8.8 | 0.11 | −0.58 |
| Poland | 0.99 | 0.67 | 0.59 |
| Russian Federation | 9.2 | −0.42 | −1.02 |
| Slovenia | 0.7 | 1.03 | 0.82 |
| South Africa | 30.8 | 0.35 | −0.16 |
| Spain | 0.8 | 1.12 | 1.06 |
| Sweden | 0.7 | 1.95 | 2.32 |
| Switzerland | 0.6 | 1.89 | 2.17 |
| Uganda | 11.5 | −0.57 | −0.98 |
| United Kingdom | 1.0 | 1.54 | 1.65 |
| United States | 4.7 | 1.52 | 1.39 |

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
