# Peer review of "The Influence of Government Effectiveness and Corruption on the High Levels of Homicide in Latin America"

_socsci, doi:10.3390/socsci10050172_

Round 1

Reviewer 1 Report

Dear author(s),

Your paper was well-written and organized, and I thoroughly enjoyed reading it. It also makes interesting arguments regarding the high levels of violence in Latin America, which indeed are a major subject in international crime trends.

I propose several suggestions on how I believe the paper could be improved. Please take my comments simply as the suggestions they are.

GENERAL COMMENTS:

  1. I really enjoyed your section 2.1. Factors explaining high homicide levels – a Latin America focus. You present several possible explanations for the violence rates of Latin America, while speaking directly about the region’s context using current data.
    1. (p. 5) However, I felt that your theoretical link between corruption and homicide was underdeveloped in this section. I suggest you include one more paragraph supporting your hypothesis.

  1. I am not sure this is related to the style of the journal, but I missed a description (even if brief) of your measures and data sources before your results section. A quick description would suffice, but I believe your analysis requires one. Otherwise, I am not sure how to interpret your associations when reading the results.
    1. I realize you include those at the end. I believe they should come sooner.

  1. Your methodological choice is not the most appropriate for your research. You propose to investigate why Latin America is not participating in an international homicide decline which is occurring over time. However, you present an analytical model which is essentially cross-sectional. Note that your model is able to account for the factors that explain the difference in violence between countries, but not to the longitudinal drivers of within-country violence over time.
    1. Hence, your paper is one explanation for why Latin America is particularly violent relative to other regions. That is a good contribution. That said, you are not explaining why the Region is experiencing a certain trajectory in its levels of violence.
    2. I believe you have two choices: either limit your research framing to focus on cross-sectional differences (in line with your method), or expand your method to include a longitudinal assessment. I believe both solutions are appropriate.
      1. I would suggest the first solution. Investigate why Latin America is so violent, not why Latin America is not experiencing a certain longitudinal change.

  1. One of the main studies to investigate the international crime decline, and you to propose an analogous explanation for the lack of a reduction in Latin America is the following.

Santos, M., Testa, A., Porter, L. C., & Lynch, J. P. (2019). The contribution of age structure to the international homicide decline. PLOS ONE, 14(10), e0222996. https://doi.org/10.1371/journal.pone.0222996

That paper speaks directly about the driver of the international homicide, as well as about the lack of a reduction in Latin America. Your study is a direct contribution to this research. Please also note the longitudinal nature of the analysis.

SPECIFIC COMMENTS:

  1. (p. 1) About the sentence: “Almost all the countries in the region have experienced a significant or moderate increase in homicides or failed to sustain reductions in homicide.” This is not entirely correct, and there are some remarkable cases in the region. Colombia is an example, recent years in Brazil, homicides in Honduras used to be even higher. Consider removing or toning down the sentence.

  1. (p. 1) About the sentence: “Several theoretical frameworks have been examined and suggested in an attempt to explain the factors that are responsible for 34 the high levels of homicide in Latin America, with most studies focusing on social inequality, poverty, unemployment, and poor levels of education .” It has an extra space in the end, and it is missing citations.

  1. (p. 4) About the sentence: “To date most cross-national studies that have examined variations in homicides have only included a small representation of countries from Latin America.” I am not sure I agree. In fact, Latin America has surprisingly high-quality homicide statistics, which cover virtually the entire region. Most models focusing on global crimes do represent Latin America. See a description at:

United Nations Office on Drugs and Crime. (2019). Global Study on Homicides 2019. Vienna, Austria: Author.

  1. (p. 6) I am also not sure what is the benefit of including that standalone indictor for Latin America. I suggest you elaborate on the purpose of that analytical decision.

  1. The two plots are fascinating, and very convincing. In fact, I believe they are as important as your model. Consider including the plots before your regression table.

Author Response

Response to reviewer in uploaded file.

Reviewer 2 Report

This is a welcome empirical study between government measures and crime. In technical terms it does not seem to work to put both government effectiveness and control of corruption in the same model due to their correlation ans subsequent inflation in the value inflation facotr (VIF). In theoretical terms a stronger case should be made to highlight the causal nature of relationship between government actvity and crime. There is also some global work done between government and corruption in gothenburg quality of government project and institute. The work has been decribed by Dahlström and Lapuente (2017) organising leviathan and in some articles. By including some of their ideas the conclusions can be strengthened.

Author Response

Response to reviewer in uploaded file.

Reviewer 3 Report

- A theoretical review is very weak. A paper of Social Sciences should have a section for theoretical discussion.

- Regression tables do not provide even basic information such as the number of observations (N). 

- VIF's of government effectiveness and control of corruption are 13 and 12. Multicollinearity issue is so serious that the model doesn't work for OLS. 

- Some values of the correlation matrix are over 1. How can they have this kind of value?

- I don't think this manuscript is ready for submission. Among many weaknesses, the flaws are obvious for rejection. 

Author Response

Response to reviewer in uploaded file.

Round 2

Reviewer 1 Report

Dear author(s),

I believe you have satisfactorily addressed my main concerns. Though I still have reservations with parts of your argument and methods, I would consider those to be intellectual differences, and not limitations of the study itself.

I appreciated the reframing of the emphasis on the effectiveness of government institutions, and the increased depth of those arguments. I also noted that you opted to maintain your original analysis, but to reframe your emphasis on factors explaining the high levels of violence in Latin America.

Overall, I found that this reframing was done well, and the article introduces compelling explanations for the high levels of violence in Latin America.

Author Response

We thank the reviewer for taking the time to review our manuscript and for their comments that have resulted in us improving the manuscript.

The reviewer stated we had satisfactorily addressed requests for revisions and did not request any further changes to the manuscript.

Reviewer 2 Report

Thank you for your amendments. There are still some corrections to be made. In table 2 there are correlations of all the variables. It suffices to show correlation between independent variables.Inclusion of dummy variable seem unnecessary. First in strains the mode with limited number of observation and second, it can be explained in qualitative part without putting it in the model. as the variables correlate it does not make sense to include both figures 1 and 2. It suffices to include either one.

Author Response

We thank the reviewer for their time in reviewing the manuscript and their comments that have resulted in us improving the manuscript.

In response to their second review:

  1. There are still some corrections to be made. In table 2 there are correlations of all the variables. It suffices to show correlation between independent variables.

Reply 2.1. We have removed the row and column for homicide so the matrix only shows independent variables.

  1. Inclusion of dummy variable seem unnecessary. First in strains the mode with limited number of observation and second, it can be explained in qualitative part without putting it in the model

Reply 2.2. We believe the inclusion of the dummy variable for Latin America is useful. First, the focus of the paper is to examine if the variables for explaining homicide variation are in some way different for the Latin American region so it makes sense to examine statistically if this is the case. Second, the statistical analysis builds on the qualitative analysis so it seems logical to compare the model results when the Latin American region is included as a dummy variable. Third, by doing so we show the model results are different – for example, we show that income inequality was no longer significant in models 3 and 4 when the dummy variable was used. And forth, the model 3 and 4 results show there to be a significant Latin America regional effect associated with government effectiveness and control for corruption and each variables’ association with homicide. We prefer to retain the models that use the dummy variable for the Latin American region. Other reviewers also found its inclusion useful.

  1. As the variables correlate it does not make sense to include both figures 1 and 2. It suffices to include either one

Reply 2.3. Another reviewer commented on the value of including both figures 1 and 2. Although government effectiveness and control for corruption are correlated, the inclusion of each figure helps to illustrate the influencing value of each variable and its relationships with homicide. Each figure also illustrates country heterogeneity within the Latin American region (a request another reviewer wanted to ensure was highlighted) but that the overall relationship for countries in the region remains (e.g. we describe that in Mexico the homicide rate was 21.3 and the control for corruption value was -0.41 compared to Chile where the homicide rate was 2.5 and the control for corruption value was 1.57.) We prefer to retain both figures for these reasons.

Reviewer 3 Report

I still think the value of this paper is not high enough to be published in a peer-reviewed academic journal. The relationships among variables do not reflect real causality. In addition, the high correlation seems taken for granted. The regression-based analysis does not use panel data. The cross-sectional data analysis is based on only 39 observations, because the number of Latin American countries in this analysis is 39. For a cross-sectional regression, I don't think the number itself makes reliable results. A multicollinearity issue is so serious. The existence of multicolllinearity makes the high level of model fit (R square value). The global indicator of government effectiveness does not reflect the government capacity for anti-crime. This paper conducts a statistical examination, but the examination does not go with strong theoretical arguments.  

Author Response

We thank the reviewer for taking the time to review our manuscript and providing further comments. In reply to the second set of comments.

  1. Theoretical aspects of the paper and in response to: "The relationships among variables do not reflect real causality"; "The global indicator of government effectiveness does not reflect the government capacity for anti-crime. This paper conducts a statistical examination, but the examination does not go with strong theoretical arguments."

Reply 2.1. Since our first manuscript we have strengthened the theoretical perspectives that we describe in the paper that could explain the presence of a relationship between institutional effectiveness and homicide rates. We describe in the 'qualitative analysis' section why institutional effectiveness is a variable that may be related with homicide levels, revisit these theoretical principles in the discussion section, and cite the work of several other researchers from a number of different disciplines (e.g., political science and criminology) who have begun to examine the influence of government effectiveness and corruption on international variations in social conditions, and how institutional weakness can have far-reaching implications in society. We recognise that other researchers may have a difference in intellectual opinion to what we state, but we think our theoretical discussion is adequate for what is still a fairly embryonic concept for helping to explain variations in homicide. We recognise and state there are limitations with the data and methods used (such as the use of the government effectiveness variable), and suggest further research that could build on our study.

Also, the statistical examination is only one part of the study. The qualitative analysis we provide argues what we believe is a compelling case for examining alternatives to structural variables for explaining variations in homicide, and is detailed in stating why institutional effectiveness warrants examination. We then conduct a modest statistical analysis to examine the presence of this relationship.   

2. Our statistical analysis, and in response to: "The regression-based analysis does not use panel data. The cross-sectional data analysis is based on only 39 observations, because the number of Latin American countries in this analysis is 39. For a cross-sectional regression, I don't think the number itself makes reliable results.", and "A multicollinearity issue is so serious."

Reply 2.2. We caveat throughout that our statistical analysis is modest and has limitations that we describe in the limitations section. We state our reasoning for conducting a cross-sectional analysis and using the sample of countries that we include in the analysis (as the starting point to examine and introduce the idea that institutional effectiveness may be related to homicide levels), and we state in the discussion section that the next step in further testing our findings could involve a panel regression analysis, involving more countries. As our focus in the manuscript is on Latin America countries we are naturally constrained by the number of countries in the region, but we believe that the findings from our modest statistical analysis and the qualitative analysis results indicate that the direction of research we have taken suggests we (and others who are also examining this as well) could be onto something that  builds on the analysis of structural variables that has come before.

In the new version of the paper we are more explicit in the results section that our VIF results for the independent variables in models 1-4 were less than five. We had previously commented on the VIF results in the methods and materials section that appears at the end of the manuscript (in accordance with the journals required manuscript structure), but this may not have been clear.

Round 3

Reviewer 2 Report

I still do not see the point in includinf both figures. The country specific differences can be explained in the text. Otherwise, this is ok. language proofing would hel to crystallise the ideas.

Reviewer 3 Report

I believe this revised version reflects substantial efforts to respond to review comments. However, as I already addressed the fundamental weaknesses of the paper in the previous review round, the revised manuscript with the very weak causal logic cannot be modified to the substantial extent.  Analyzing the data of 39 countries cannot make any qualitatively significant finding.